# Burnout Syndrome among Pediatric Nephrologists—Report on Its Prevalence, Severity, and Predisposing Factors

**DOI:** 10.3390/medicina58030446

**Published:** 2022-03-18

**Authors:** Ewa Pawłowicz-Szlarska, Piotr Skrzypczyk, Małgorzata Stańczyk, Małgorzata Pańczyk-Tomaszewska, Michał Nowicki

**Affiliations:** 1Department of Nephrology, Hypertension and Kidney Transplantation, Medical University of Lodz, 92-213 Lodz, Poland; ewa.pawlowicz@umed.lodz.pl; 2Young Nephrologists’ Club, Polish Society of Nephrology, 02-006 Warsaw, Poland; 3Department of Pediatrics and Nephrology, Medical University of Warsaw, 02-091 Warsaw, Poland; pskrzypczyk@wum.edu.pl (P.S.); malgorzata.panczyk-tomaszewska@wum.edu.pl (M.P.-T.); 4Young Nephrologists’ Forum, Polish Society of Pediatric Nephrology, 93-338 Lodz, Poland; malgorzata.barbara.stanczyk@umed.lodz.pl; 5Department of Pediatrics, Immunology and Nephrology, Polish Mother’s Memorial Hospital Research Institute, 93-338 Lodz, Poland

**Keywords:** burnout, occupational phenomenon, pediatric nephrology, pediatric nephrologists

## Abstract

Background and Objectives: Burnout is an occupation-related syndrome comprising emotional exhaustion, depersonalization, and reduced feelings of work-related personal accomplishments. There are reports on burnout among adult nephrologists and general pediatricians, but little is known about burnout among pediatric nephrologists. The aim of our study was to assess the prevalence and severity of burnout syndrome among Polish pediatric nephrologists. Materials and Methods: A 25-item study survey consisting of abbreviated Maslach Burnout Inventory and additional self-created questions about work-related factors was completed by 97 physicians affiliated with the Polish Society of Pediatric Nephrology. Women comprised 75.3%, with median time of professional experience in the study group was 15 years. Results: A high level of emotional exhaustion, depersonalization, and reduced feeling of personal accomplishments were observed in 39.2%, 38.1%, and 21.6% of the participants, respectively. At least a medium level of burnout in all three dimensions were observed in 26.8% of the participants and 8.2% of them presented high three-dimensional burnout. About 41.2% of the participants stated that they would like to take part in burnout prevention and support programs. According to the study participants, excessive bureaucracy in healthcare systems, rush at work, and overtime work were the main job-related problems that could influence burnout intensity. Conclusions: Burnout is an important factor in the professional landscape of pediatric nephrology. Actions aimed at reducing the risk of occupational burnout among pediatric nephrologists should be applied, both at the personal and institutional levels.

## 1. Introduction

Burnout was first described in 1974 by Herbert Freudenberger, who observed emotional distress and chronic fatigue among volunteer workers at free clinics for drug addicts in New York City. He defined burnout as a “state of mental and physical exhaustion caused by one’s professional life” [1]. Further research on this phenomenon, led by American social psychologist Christina Maslach, has yielded the most commonly used definition of burnout, described as an occupation-related syndrome comprising emotional exhaustion, depersonalization, and reduced feelings of work-related personal accomplishments [2]. In 1981, Maslach and colleagues created the first psychological tool for burnout measurement—the Maslach Burnout Inventory [3], which more than 30 years later is still referred to as the main measure of occupational burnout [4]. Burnout was included in the 11th Revision of the International Classification of Diseases (ICD-11) as an occupational phenomenon. However, it is not classified as a medical condition but as a factor affecting health status or contact with health services [5].

Recently, occupational burnout has been gaining importance as a significant threat to professional life in the healthcare. It has been confirmed that physicians are at a greater risk compared to all other professions in the US. Some medical specialties are considered more at risk of burnout, e.g., family medicine, general internal medicine, and emergency medicine [6]. However, burnout should be measured and assessed in many different areas of medicine, taking into account the specificities of these specialties, which can, in particular, affect work-related conditions and increase burnout rates. For adult nephrology, many specific factors have been defined that influence burnout in this population of physicians—for example, long-term relationships being established between the caregiver and the patient, as well as the chronic character of most kidney diseases [7]. Additionally, we recently confirmed that working in dialysis settings is associated with higher rates of reduced personal accomplishment, compared to working in nephrology departments in hospitals [8]. This may be associated with specific dialysis therapy-related factors, i.e., poor prognosis of patients in chronic hemodialysis, significant comorbidities in this population, and poor adherence to medical recommendations, especially with regard to dietary and fluid restrictions [9,10,11].

Burnout among pediatricians has already been described in several reports, addressing this phenomenon mostly in the population of general pediatricians and in some sub-areas, i.e., in physicians working in neonatal intensive care units and children’s emergency departments [12,13].

In 2019, the American Society of Pediatric Nephrology (ASPN) convened a national Workforce Summit to address challenges faced by the pediatric nephrology population. The conference gathered a diverse group of pediatric nephrologists with interest and expertise in workforce development, who addressed four main groups of challenges, namely: (1) Trainee Education and Exposure; (2) Reimbursement and Policy Issues; (3) Fellowship Training Duration; and (4) Retention, Burnout, and Professional Well-Being. In the summary of the discussion on the Retention, Burnout and Professional Well-Being group, it was concluded that addressing and mitigating burnout is essential for not only a thriving workforce but also in avoiding the consequences of burnout, such as high turnover, patient dissatisfaction, and a lower quality care [14].

However, data on the prevalence, severity, and characteristics of burnout syndrome in the population of pediatric nephrologists are lacking.

Pediatric nephrology is characterized by some unique features among other medical specialties. Firstly, pediatric nephrologists work in a niche area of medicine, and the vast majority of diseases that they tackle meet the European criteria for rare diseases. Furthermore, pediatric patients with kidney diseases are characterized by numerous co-morbidities, e.g., affecting the nervous and cardiovascular systems and a vast range of symptoms (short stature, malnutrition, etc.). Not infrequently, in highly burdened patients, renal survival is the limiting factor of their survival. It is, thus, necessary to work together in a multispecialty team using advanced equipment, such as a continuous renal unit connected to extracorporeal membrane oxygenation. All of these create a great workload and highly specific professional landscape of pediatric nephrology. Therefore, burnout among pediatric nephrologists should be assessed separately from other pediatricians and adult nephrologists.

Taking into account the lack of data on burnout in pediatric nephrologists, a joint project of the Young Nephrologists’ Forum of the Polish Society of Pediatric Nephrology (PTNefD) and the Young Nephrologists’ Club of the Polish Society of Nephrology (PTN) has been established to assess this phenomenon in the population of pediatric nephrologists. The results of the study can form the basis for further preventive and corrective actions in affected physicians.

## 2. Materials and Methods

### 2.1. Study Survey

Invitations to take part in a web-based survey created using the Survey Monkey application were sent by e-mail to all members of the Polish Society of Pediatric Nephrology. The study was conducted from 15 January 2019 to 20 February 2020. In the middle of this period, reminder e-mails were resent.

The 25-item study survey consisted of abbreviated Maslach Burnout Inventory with additional 16 self-created questions about work-related factors, i.e., years of professional service, place of work, holiday leave, as well as self-assessment of the causes of burnout and the strategies used by respondents to deal with burnout symptoms. Demographic data collected through the survey included only age and gender (Appendix A). The approximate time to complete the survey was expected to be 5–10 min.

The abbreviated Maslach Burnout Inventory (aMBI) is a 9-item validated survey, derived from the original 22-item Maslach Burnout Inventory Human Service Scales (MBI-HSS) [15,16]. The survey was translated into Polish by an experienced translator with vast experience in medical language and additionally verified by clinicians. aMBI comprises three statements referring to each one of the three dimensions of burnout, i.e., emotional exhaustion, depersonalization, and personal accomplishment. Each statement was rated by respondents on a timescale where “never” indicated 0 points and “daily” 6 points. The sum of the points in particular dimensions allowed qualifying burnout intensity as low, moderate, and high using the following intervals: personal accomplishment: <13 high, 13–14 moderate, >14 low burnout; depersonalization: >6 high, 4–6 moderate, <4 low burnout and emotional exhaustion: >10 high, 7–10 moderate, <7 low burnout [16]. There is marked variation in burnout definitions and the importance of developing consensus around this is highlighted [17]. We present results as burnout intensity using the abovementioned score intervals and refer to three burnout subcomponents in all analyses.

The introductory letter to the survey contained a statement that by completing the survey, the respondent consented to participating in the survey. It also stated that the survey was completely anonymous. The rationale for the project was clearly described.

### 2.2. Study Group

The study group comprised 97 physicians affiliated with the Polish Society of Pediatric Nephrology, which accounts for 43% of the society’s members.

The only exclusion criterion was work outside pediatric nephrology settings. Residents, specialists in general pediatrics training for pediatric nephrology, and specialists in pediatric nephrology were enrolled.

### 2.3. Statistical Analysis

Statistical analysis was performed using Statistica version 13.1 PL software, Dell Inc., Tulsa, OK, USA. Graphs were plotted with MS Excel, and word cloud was created with free online application TagCrowd (https://tagcrowd.com/, accessed on 29 August 2021). The distribution of continuous variables was assessed with the Shapiro–Wilk test. The Mann–Whitney U test was used for comparisons between two independent groups. Non-parametric comparisons of more than two groups were performed with the Kruskal–Wallis test. To determine internal consistency reliability, Cronbach alpha was calculated for each dimension (3 items per dimension) of the Maslach Burnout Inventory. It amounted to 0.84, 0.82, and 0.62 for emotional exhaustion, depersonalization, and reduced personal accomplishment, respectively.

## 3. Results

The characteristics of the study population are provided in Table 1, the study dataset is provided in Appendix A. The actual mean time to complete the survey was 4 min 55 s.

Detailed results of the aMBI in the study group are provided in Table 2. Percentage of the study group presenting low, medium, or high level of burnout in all dimensions is provided in Table 3. It was seen that 26.8% (*N* = 26) of participants presented at least medium level of burnout in all three dimensions but only 8.2% of them (*N* = 8) presented high three-dimensional burnout. No differences in burnout intensity in any of the three dimensions were found between female and male pediatric nephrologists. Interestingly, job seniority also did not correlate with burnout intensity. The use of holiday leave did not influence burnout rates. Since there were only three respondents working mostly in dialysis units, a comparison of burnout between hospital-based and dialysis-based pediatric nephrologists was not possible.

The participants were asked if they felt burned out, to which 35.1% (*N* = 34) answered “yes” or “rather yes”, 45.4% stated “rather no”, while 19.5% definitely denied being burned out. Self-assessed burnout was significantly associated with burnout measured with aMBI in all three dimensions (emotional exhaustion *p* = 0.0002, depersonalization *p* < 0.0001, reduced personal accomplishment *p* < 0.0001; Kruskal–Wallis test).

Only one person in the study group reported current participation in burnout prevention and/or support programs, while 41.2% of participants stated that they would like to take part in such programs. Asked about preferences regarding prevention and support programs, 50% of those interested in such activities stated that they preferred individual meetings with psychologist and the other half favored group meetings with colleagues suffering from burnout. It was revealed that 70% of them felt such programs should be voluntary and employer-funded, while 30% stated that participation in support programs should be obligatory for those diagnosed with burnout symptoms.

Of the respondents, 46.4% (*N* = 45) said they undertook actions of their own in order to reduce burnout; 37.1% considered meetings with family/friends as a burnout remedial measure, 29.9% played sports, 19.6% preferred to sleep, and 15.5% listened to music.

Job-related issues that may influence burnout intensity and the number of participants who stated that they could contribute to burnouts are provided in Figure 1. Among other issues worsening the working environment and promoting burnout, the participants mentioned difficulties in contact and cooperation with patients’ parents, lack of organizational and personal support from hospital and university authorities, no access to the most modern treatment methods for economic reasons, transferring responsibility for accounting procedures to doctors, and repeatability of work.

## 4. Discussion

Our results indicate that burnout should be considered an important factor in the professional landscape of pediatric nephrology. More than one in four physicians in the Polish pediatric nephrology population suffer from burnout of at least a medium level. However, lack of data on this phenomenon points to the underestimation of its meaning for patients and healthcare professionals.

It has been confirmed that intension to leave the profession, career ending, early retirement [18], and high turnover [19] are related to burnout in healthcare. These challenges are also faced by pediatric nephrology professionals [20]. A complex description of the US pediatric nephrology workforce was provided in 2015 in a report commissioned by the American Academy of Pediatrics. Among physicians who planned to cut back on clinical activities for reasons other than retirement, dissatisfaction with work–life balance was the leading cause [21]. Even though burnout-related issues were not addressed in the report, it might be considered as one of the causes for reducing clinical activities or even leaving the subspecialty. An assessment of burnout and creating burnout support programs may contribute to reduction of these unfavorable phenomena.

The only report we found specifically addressing burnout in the pediatric nephrology population was the poster presentation at the Educational Research On-demand session at ASN Kidney Week 2020. In the report entitled The Sustainable Pediatric Nephrology Workforce Project (SUPERPOWER): A Pilot Study of Burnout and Resilience, its authors presented data on burnout, resilience, sleep, stress, and self-compassion gathered from 116 pediatric nephrologists (30 fellows and 86 faculty) from 11 US pediatric nephrology programs. Personal characteristics associated with burnout included lower resilience, higher stress, and sleepiness. It is of note that, similar to our results, demographic characteristics were not associated with burnout in the US cohort. The authors concluded that burnout among pediatric nephrologists was below the average reported by US physicians overall [22].

Kemper et al., on behalf of the Pediatric Resident Burnout-Resilience Study Consortium, assessed the epidemiology of burnout among US pediatric residents. It was concluded that more than 50% of pediatric residents met the criteria for burnout. Burnout rates were not associated with any demographic or residency characteristics. Similarly, in our study, we were not able to find a direct association between demographic characteristics and job-related parameters. According to Kemper et al., factors associated with an increased risk of burnout were stress, sleepiness, dissatisfaction with work–life balance, and recent medical errors [23].

Although nephrologists and pediatric nephrologists share many similarities in work, it is important to emphasize the specificity of the work of physicians who treat kidney disease in children. Pediatric patients with kidney disease are characterized by a wide variety of diseases and manifestations. Factors increasing workload include increased comorbidity, e.g., malformations, developmental delay, and recurrence of many diseases (e.g., idiopathic nephrotic syndrome). Unlike in adult nephrology, the number of dialysis patients at each center is small. Nevertheless, a comparison of burnout between pediatric and adult nephrologists could be interesting.

Our recent study on burnout in Polish adult nephrologists was the first attempt to characterize the phenomenon, its prevalence, and severity [8]. This study showed that pediatric nephrologists presented lower burnout rates in all dimensions, as compared to adult nephrologists. High levels of emotional exhaustion, depersonalization, and reduced personal accomplishment were observed in 49.1%, 52%, and 31.4% of adult nephrologists and 39.2%, 38.1%, and 21.6% in pediatric nephrologists, respectively. At the bare minimum, a medium level of burnout in all three dimensions was found in 41.8% adult and 26.8% pediatric nephrologists. Depersonalization and reduced feelings of personal accomplishment were significantly less pronounced in pediatric nephrologists. It may be hypothesized that better prognosis of some diseases, e.g., glomerulopathies, more patients treated by peritoneal dialysis than hemodialysis, and a much higher rate of those patients with end-stage kidney disease receiving kidney transplant in the pediatric compared to the adult population, may make greater sense of personal accomplishments among pediatric nephrologists; however, further studies, ideally of qualitative methodology, are needed to verify this. Interestingly, significantly more adult nephrologists (62.8%) than pediatric nephrologists (41.2%) would like to take part in burnout prevention and/or support programs.

The same psychometric instrument was used in both populations; however, these comparisons may be affected by the different way the data were collected in these two studies—a paper survey for adult nephrologists and an on-line survey for pediatric nephrologists.

One of the limitations of our study is the small study group. According to the National Chamber of Physicians 31.12.2021, there were 114 board-certified pediatric nephrologists in Poland [24]—42.9% of them (49 pediatric nephrologists in our study group), along with 48 general pediatricians and residents working in pediatric nephrology settings seem to be enough to draw conclusions about this population. Besides, the only other report that focused specifically on burnout among pediatric nephrologists in the US comprised only 116 participants, and the number of pediatric nephrologists in the US is about 500 [22,25]. Besides, the social-desirability bias, understood as a tendency of survey respondents to answer questions in a way that they are viewed favorably by others, must be considered in our study [26]. It could especially play a role in answering questions regarding the depersonalization dimension of burnout.

The use of the abbreviated version of the gold-standard instrument (Maslach Burnout Inventory, Mind Garden, Inc., CA, USA) could be mentioned as a limitation of our study. However, it has been proven that aMBI may accurately assess burnout in medical professionals and that three-item aMBI subscales are valid and reliable proxies of MBI-HSS counterparts [15].

## 5. Conclusions

Burnout is an important, yet heavily underestimated problem in pediatric nephrology. More than one-fourth of Polish pediatric nephrologists suffer from burnout on at least an average level. Lack of direct associations between demographic characteristics or job-related parameters and burnout should motivate further research in the area. Specialty-specific factors, along with personal and institutional parameters, should be addressed to decrease the risk of burnout in the population of pediatric nephrologists.

## Figures and Tables

**Figure 1 medicina-58-00446-f001:**
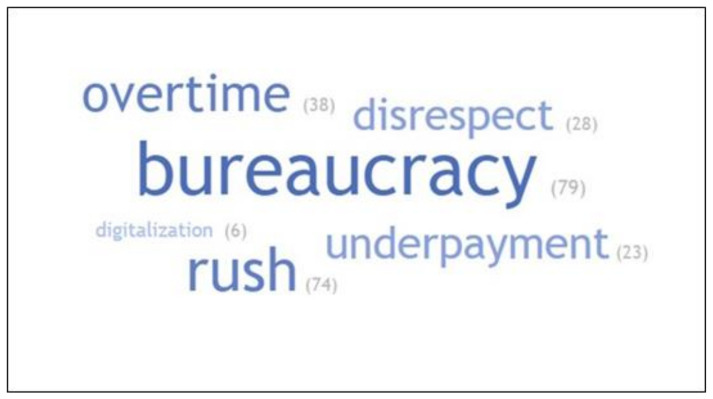
A word cloud presenting job-related issues influencing burnout intensity among pediatric nephrologists. The more a specific factor appeared in the database, the bigger and bolder it appears in the word cloud.

**Table 1 medicina-58-00446-t001:** The characteristics of the group of pediatric nephrologists in Poland.

Characteristic	Number of Participants (%) or Median (IQR, Min–Max)
Gender	
Men	21 (21.6%)
Women	73 (75.3%)
No information	3 (3.1%)
Age	
<30 years	10 (10.3%)
30–50 years	54 (55.7%)
51–62 years	29 (29.9%)
>65 years	1 (1%)
No data	3 (3.1%)
Years of professional experience [median (IQR, min–max)]	15 (20.5, 1–40)
Level of professional experience	
Board-certified pediatric nephrologist	49 (50.5%)
Board-certified general pediatrician	17 (17.5%)
Residents	21 (21.7%)
No information	10 (10.3%)
Main workplace	
Hospital (pediatric nephrology department)	81 (83.5%)
Hospital (other department)	6 (6.2%)
Nephrology out-patient clinic	1 (1%)
Dialysis unit	3 (3.1%)
No information	6 (6.2%)
Number of workplaces	
1	13 (13.4%)
2	48 (49.5%)
3 and more	30 (30.9%)
No information	6 (6.2%)
Working hours per week	
<40	4 (4.1%)
41–50	37 (38.2%)
51–60	29 (29.9%)
61–75	13 (13.4%)
>75	8 (8.2%)
No information	6 (6.2%)
Use of holiday leave	
Fully	53 (54.6%)
Partially	37 (38.2%)
No information	7 (7.2%)

IQR—interquartile range.

**Table 2 medicina-58-00446-t002:** Results of the abbreviated Maslach Burnout Inventory in the group of pediatric nephrologists.

Burnout Dimensions and Items	*N*	Never	A Few Times a Year	Once a Month or Less	A Few Times a Month	Once a Week	A Few Times a Week	Every Day
Emotional exhaustion								
(1) Feeling emotionally drained from work.	97	7.2%	16.5%	11.3%	18.6%	15.5%	24.7%	6.2%
(2) Feeling fatigue when facing another day at work.	97	7.2%	15.5%	17.5%	17.5%	8.2%	18.6%	15.5%
(3) Feeling that working all day with people is a strain.	97	17.5%	20.6%	14.4%	9.3%	10.3%	20.6%	7.2%
Depersonalization								
(1) Treating some patients as impersonal objects.	97	27.8%	18.6%	7.2%	18.6%	10.3%	16.5%	1.0%
(2) Becoming more callous towards people since taking this job.	97	32.0%	14.4%	11.3%	12.4%	11.3%	15.5%	3.1%
(3) Not really caring what happens to some patients.	97	52.6%	20.6%	12.4%	1.0%	6.2%	7.2%	0.0%
Personal accomplishment								
(1) Dealing very effectively with patient problems.	96	0.0%	1.0%	0.0%	0.0%	5.2%	41.7%	52.1%
(2) Feeling like they have a positive influence on other people’s lives.	97	3.1%	4.1%	4.1%	8.2%	14.4%	37.1%	28.9%
(3) Feeling exhilarated after working closely with patients.	96	1.0%	2.1%	5.2%	15.6%	11.5%	39.6%	25.0%

**Table 3 medicina-58-00446-t003:** Prevalence of high, moderate, and low levels of burnout in each dimension.

Burnout Level in Particular Dimension	% of Respondents
Emotional exhaustion	
Low	34.0%
Medium	26.8%
High	39.2%
Depersonalization	
Low	43.3%
Medium	18.6%
High	38.1%
Reduced personal accomplishment	
Low	58.8%
Medium	19.6%
High	21.6%

## Data Availability

The manuscript has data included as electronic Appendix A.

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
