# Peer review of "Burnout Syndrome among Pediatric Nephrologists—Report on Its Prevalence, Severity, and Predisposing Factors"

_medicina, 2022, doi:10.3390/medicina58030446_

Round 1

Reviewer 1 Report

The manuscript is written clearly but rather long. It deserves publication.  I have a few concerns.

Methods

  1. The abbreviated MBI (aMBI) is used to diagnose burnout. Please provide a reference to proof your statement that the aMBI is a validated survey.
  2. Was the Polish version validated?
  3. Please explain how burnout is diagnosed. You should detail this in the methods section. This is not a simple matter. You might refer to Rotenstein 2018 in JAMA. This will probably also have consequences for the way you present the results.
  4. You might consider putting Table 1 in the results section. After all, it shows results.

Results

  1. May I suggest a table with results comparing personal and work characteristics on the one hand with either burnout or no burnout on the other hand? This could replace table 2, which is not very interesting.
  2. How was table 3 made? That is: how did you calculate 34.0% for low emotional exhaustion, etc? Please describe this in the methods.

Discussion

  1. The discussion is rather long. May I suggest to focus more on your own results and make it more concise? From line 173 to line 224 you discuss other studies not referening to the present study. Starting from line 224 you compare results with a previous study in adult nephrologists. This is interesting but speculative. First of all, are results comparable and are the differences statistically significant? Subsequently you elaborate on possible causes, but are these hypotheses based on your data? I do not think so.
  2. What makes pediatric nephrologists so special they have to be studies separately? (this could also be placed in the introduction)
  3. Please reflect on the methods, I mean on the fact that you asked participants if they were burnout and why. Do you think asking for burnout might have influenced the results?
  4. You state in line 243 that 97 out of 114 board certified pediatric nephrologist participated but in line 111 you state that 97 is 43% of the members…?

Author Response

Dear Reviewer,

We appreciate very much the time and effort that you have dedicated to providing your valuable feedback on our manuscript. We are very grateful for your insightful comments on our paper. Please find below our responses to your criticism. All changes within the revised version of our manuscript have been highlighted.

Yours sincerely,

Ewa Pawłowicz-Szlarska

Responses to Reviewer’s comments

General remarks:

1) English language and style are fine/minor spell check required

Thank you for this suggestions. The revised version of the manuscript was checked for English grammar and spelling by a native speaker.

Methods:

1) The abbreviated MBI (aMBI) is used to diagnose burnout. Please provide a reference to proof your statement that the aMBI is a validated survey.

Thank you very much for this suggestion. We have added also in the Methods section the reference (Riley et al.), which was used before only in the Discussion section.

2) Was the Polish version validated?

This version of the inventory was not validated specifically in the group of Polish physicians. It was, however, translated by an experienced translator with the vast experience in medical language and additionally verified by clinicians. We have added this information in the Methods section. The same instrument was used in our previous publication (Pawlowicz et Nowicki, 2020).

3) Please explain how burnout is diagnosed. You should detail this in the methods section. This is not a simple matter. You might refer to Rotenstein 2018 in JAMA. This will probably also have consequences for the way you present the results.

Thank you for this important remark. We have added the paragraph in the Methods section referring to the abovementioned paper, namely: “There is marked variation in burnout definitions and the importance of developing consensus around this is highlighted [17]. We present results as burnout intensity using the abovementioned score intervals and refer to three burnout subcomponents in all analyses.”

4) You might consider putting Table 1 in the results section. After all, it shows results.

Thank you for this remark. According to your suggestion the Table 1. was moved to the Results section.

Results:

1) May I suggest a table with results comparing personal and work characteristics on the one hand with either burnout or no burnout on the other hand? This could replace table 2, which is not very interesting.

As we reported in the Results section “No differences in burnout intensity in any of the three dimensions were found between female and male pediatric nephrologists. Interestingly, also job seniority did not correlate with burnout intensity. The use of holiday leave did not influence burnout rates.”, so we did not decide to present this in a separate table. Our results in this regard are convergent with the US cohort of the SUPERPOWER project (Halbach, 2020). Since there are issues around conclusive definition of burnout (Rotenstein, 2018), we wanted to present the raw results of the aMBI (Table 2). In Table 3 we present burnout results in more informative manner.

2) How was table 3 made? That is: how did you calculate 34.0% for low emotional exhaustion, etc? Please describe this in the methods.

Thank you very much for this question. We have added the paragraph with the appropriate reference in the Methods section, in which we clearly described the way to calculate burnout intensity. It was calculated as follows: “Sum of points in particular dimensions allowed qualifying burnout intensity as low, moderate and high using the following intervals: personal accomplishment: < 13 high, 13–14 moderate, > 14 low burnout; depersonalization: > 6 high, 4–6 moderate, < 4 low burnout and emotional exhaustion: > 10 high, 7–10 moderate, < 7 low burnout.”

Discussion:

1) The discussion is rather long. May I suggest to focus more on your own results and make it more concise? From line 173 to line 224 you discuss other studies not refering to the present study. Starting from line 224 you compare results with a previous study in adult nephrologists. This is interesting but speculative. First of all, are results comparable and are the differences statistically significant? Subsequently you elaborate on possible causes, but are these hypotheses based on your data? I do not think so.

Thank you very much for this suggestion. In line with this remark we have significantly shortened the paragraph, leaving the comments on the only study apart from ours, which addressed specifically burnout in pediatric nephrologists SUPERPOWER PROJECT (Halbach, 2020).

As for the second part of this comment - the comparison between pediatric and adult nephrologists was possible, since both studies used the same instrument (aMBI), the only bias could be the method of data gathering (paper surveys vs. online survey) what was disclosed in this paragraph. Presented differences are statistically significant. Our previous publication (Pawlowicz et Nowicki, 2020) reported on burnout among adult nephrologists and we used our previous results to provide these comparisons. We have substantially corrected the paragraph, in which we indicated the potential causes and underlined a need for further, ideally qualitative, studies to verify these hypotheses.

2) What makes pediatric nephrologists so special they have to be studies separately? (this could also be placed in the introduction)

Thank you for this suggestion. We have added this paragraph in the Introduction section: “Pediatric nephrology is characterized by some unique features among other medical specialties. Firstly, pediatric nephrologists work in a niche area of medicine, and the vast majority of diseases that they tackle with meet the European criteria for rare diseases. Furthermore, the  pediatric patients with kidney diseases are characterized by numerous co-morbidities, e.g., affecting the nervous and cardiovascular systems and a vast range of symptoms (short stature, malnutrition, etc.). Not infrequently, in highly burdened patients, renal survival is the limiting factor of their survival. It is necessary to work together in the multispecialty team using an advanced equipment such as a continuous renal unit connected to extracorporeal membrane oxygenation. All of these create a great workload and highly specific professional landscape of pediatric nephrology. Therefore, burnout among pediatric nephrologists should be assessed separately from other pediatricians and adult nephrologists.”

3) Please reflect on the methods, I mean on the fact that you asked participants if they were burnout and why. Do you think asking for burnout might have influenced the results?

Thank you very much for this question. Besides using any objective measures of aMBI, we wanted above all to address self-perceived burnout among study participants. As it is mentioned in the Results section it was significantly associated with all burnout dimensions. Although it is disputable we may doubt that the fact that we asked about the self-perceived feeling of burnout could bias our results.

4) You state in line 243 that 97 out of 114 board certified pediatric nephrologist participated but in line 111 you state that 97 is 43% of the members…?

Thank you very much for this remark. The society’s members are not only board-certified specialists of pediatric nephrology but also board-certified general pediatricians and residents, who were also enrolled in our study. The text in line 243 “According to the National Chamber of Physicians 31.12.2021, there were 114 board-certified pediatric nephrologists in Poland.” was provided to show that the population of specialists in pediatric nephrology in our country is very small. Our study population is small in general but it could not be greater when we consider such a small population. In fact, later on in this part we refer to the whole group (97) and not specifically specialists (49 in our study population, as provided in Table 1). We have clarified this paragraph by adding number and percentage of board-certified specialists of pediatric nephrology.

Reviewer 2 Report

The manuscript by Pawłowicz-Szlarska et al. is a well-written paper that investigates burnout prevalence and severity in the Polish pediatric-nephrology providers. This is the current topic, and the results are intriguing, although not unexpected. In general, the manuscript is well written, informative, and easy to follow, although there are some (mostly minor) issues that need to be addressed as outlined below:

-would authors be able to add standard deviation (SD) and range for the years of experience

-Line 255: sentence end is missing

Thank you for the opportunity to review this interesting article.

Author Response

Dear Reviewer,

We appreciate the time and effort that you have dedicated to providing your valuable feedback on our manuscript. We are very grateful for your comments on our paper. Please find below our responses to your suggestions. We have highlighted the changes within the revised version of our manuscript.

Yours sincerely,

Ewa Pawłowicz-Szlarska

Responses to Reviewer’s comments

1) English language and style are fine/minor spell check required

Thank you for this suggestions. The revised version of the manuscript was checked for English grammar and spelling.

2) Would authors be able to add standard deviation (SD) and range for the years of experience.

Thank you very much for this suggestion. Since the variable time of professional experience is of non-normal distribution we did not used mean and SD but median and interquartile range. This is why we would like not to present SD here, but we have clarified what kind of data are presented and we have added range (min-max) for this variable in the table.

3) Line 255: sentence end is missing

Thank you very much for this remark. We have added the missing part of this sentence in the revised version of the manuscript.
